# Functionalization of Glucose Oxidase in Organic Solvent: Towards Direct Electrical Communication across Enzyme-Electrode Interface

**DOI:** 10.3390/bios12050335

**Published:** 2022-05-13

**Authors:** Vygailė Dudkaitė, Gintautas Bagdžiūnas

**Affiliations:** 1Group of Supramolecular Analysis, Department of Bioanalysis, Institute of Biochemistry, Life Sciences Center, Vilnius University, Saulėtekio Av. 7, LT-10257 Vilnius, Lithuania; vygaile.dudkaite@chgf.stud.vu.lt; 2Department of Functional Materials and Electronics, Center for Physical Sciences and Technology, Saulėtekio Av. 3, LT-10257 Vilnius, Lithuania

**Keywords:** glucose oxidase, ferrocene, direct electron transfer, chloroform, bioelectrocatalysis, monolayer

## Abstract

Enzymatic biosensors based on glucose oxidase has been proven to be one of the effective strategies for the detection of glucose and contributed to health improvements. Therefore, research and debates to date are ongoing in an attempt to find the most effective way to detect this analyte using this enzyme as the recognition center. The 3rd generation biosensors using direct electron transfer (DET) type enzymes are a great way towards practical devices. In this work, we developed a simple method for the functionalization of glucose oxidase with redoxable ferrocene groups in chloroform. The enzyme retained its activity after storage in this organic solvent and after the functionalization procedures. This enzyme functionalization strategy was employed to develop the biosensing monolayer-based platforms for the detection of glucose utilizing the quasi-DET mechanism. As a result of an electrochemical regeneration of the catalytic center, the formation of harmful H_2_O_2_ is minimized during enzymatic electrocatalysis.

## 1. Introduction

Glucose is the main source of energy for the human body and brain. Abnormal glucose levels in the blood can lead to diabetic disease. Therefore, the development of an efficient glucose detection system still has paramount significance [1]. Due to selectivity to glucose, electrochemical biosensors based on glucose oxidase (GOx) as a cheap and stable enzyme from *Aspergillus niger* have commonly been used to determine blood glucose levels [2]. GOx is naturally involved in the oxidation of glucose by oxygen to gluconic acid and relays hydrogen peroxide. These biosensors can be divided into three generations according to the analytical signal registration principle [1]. First-generation glucose biosensors, which are the most used at present, are based on electrochemical detection of realized hydrogen peroxide. Second-generation biosensors are based on some auxiliary redox mediators that transfer electrons from the enzyme to the electrode. The third-generation biosensors based on the direct electron transfer (DET) from the immobilized enzyme to the working electrode are the most promising technology for glucose detection. By operating this type of biosensor, hydrogen peroxide, which can degrade the enzyme on the electrode, is not released.

Researchers are still debating whether this DET mechanism is possible in the case of GOx because flavin adenine dinucleotide (FAD) as a cofactor is deeply hidden inside this enzyme and DET is likely not possible [3,4]. However, based on our theoretical investigation of the charge carrier transfer mechanisms between GOx and organic semiconductors, we found that hole hopping between the reduced cofactor and the organic semiconductor modified electrode surface through some aromatic redoxable amino acid residues is possible when the ionization potential of the semiconductor is lower than 5.2 eV vs. vacuum. Moreover, electrons from the oxidized cofactor are transferred by the long range direct tunneling mechanism and this process strongly depends on the distance between the enzyme and the electrode surface [5]. The chains of aromatic redox-able amino acids, such as tyrosine and tryptophan have recently been proposed by Zanetti-Polzi et al. to provide protection to the active site and the whole protein by delivering holes as the oxidizing equivalents out of the protein via a multistep hopping mechanism [6]. To prove these statements, the biosensors based on carbazole, and triphenylamine polymers as the hole transporting semiconductors on the electrodes with GOx were tested in our laboratory [7,8,9]. However, these biosensors have not demonstrated excellent characteristics to apply in practice due to the relatively large distance between the cofactor of GOx and the electrode surface. Therefore, functionalization of GOx with some low ionization potential ferrocene (Fc)-based compounds might solve this problem because the Fc groups will be on the surface of GOx and can work as a hole-mediator. Moreover, the Fc-based redox polymers and derivatives have often been applied for glucose sensing using various strategies [10].

Recently, chemical tools for the site-selective functionalization of proteins opened a new area for a wide range of applications of this semi-synthetic protein conjugates towards biomedicine, bioimaging, and biosensing [11]. Suzuki et al. rationally introduced a lysine residue onto the GOx surface by the genetic engineering methods in order to connect phenazine ethosulfate as a redox mediator and used this conjugate for glucose biosensors with a quasi-DET response [12]. The authors have assigned these biosensors with this response as “2.5th generation” because by definition they are between the 2nd (via mediator) and 3rd (DET) generations [13]. Moreover, Fan et al. functionalized GOx with hemin prepared nanogel and adapted these biostructures for the optical determination of glucose [14]. Functionalization of side chains of amino acids of GOx with water soluble Fc derivatives, such as ferrocene carboxylic acid [15,16] and ferrocylacetic acid [17] was used to develop modified GOx-based biosensors. Unfortunately, these conjugation methods are not suitable for reagents that are insoluble in water.

In this work, we used chloroform as an organic solvent for the functionalization of GOx with ferrocenyl aldehyde. Activities of this modified enzyme after storage in this organic solvent and after the functionalization procedures were tested and compared with data of the unaffected enzyme. We have also shown that these enzymes can be used as the active monolayers for the glucose 2.5th generation biosensors.

## 2. Materials and Methods

### 2.1. Materials and Reagents

Organic solvents were purchased from ChemPur (Piekary Śląskie, Poland) and distilled prior to the experiments. The used lyophilized powder enzymes of type VII glucose oxidase (GOx) from *Aspergillus niger* of ≥100 U mg^−^^1^ (without added oxygen) and type I peroxidase from horseradish (HRP) of approximately 150 U mg^−^^1^ were purchased from Sigma Aldrich. All inorganic salts for the buffer solutions were purchased from Carl Roth GmbH + Co. KG. Ferrocenecarboxaldehyde, 4-mercaptobenzoic acid (MBA), 1-ethyl-3-(3-dimethylaminopropyl)carbodiimide (EDC) and *N*-hydroxysuccinimide (NHS) were purchased from Sigma Aldrich. All the experiments were conducted using type II water (R > 18 MΩ) purified in a Milli-Q system; 50 mM pH 7.0 potassium phosphate buffer (PPB) solution was used.

### 2.2. Instrumentations

Electrochemistry of our samples was carried out using an EmStat3+ potentiostat PSTrace5 from PalmSens, Houten, The Netherlands. Ultraviolet–visible (UV–Vis) spectra were measured to evaluate the kinetic constants and for the structure characterization using an Evolution 300 Security UV–Vis Spectrophotometer (Thermo Fisher Scientific, Waltham, MA, USA). During the measurements, the samples were thermostated at 25 °C using an integrated thermostat. A JASCO J-815 Circular Dichroism spectrometer (JASCO, Tokyo, Japan) was used to measure the circular dichroism (CD) spectra of the modified and native GOx in a buffer of pH 7.0. These CD spectra were carried out under a stream of Ar gas (5 L min^−1^) and in a wavelength range of 190–500 nm with a scan rate of 50 nm min^−1^ in a cell of 0.10 cm path length, the concentration of GOx solutions was 1.0 mg mL^−1^. The CD spectra were recorded two times and the results were averaged. Hydrodynamic diameters of our samples were measured using Zetasizer µV (Malvern Panalytical Ltd, Great Malvern, United Kingdom) and its compatible software. Prior to the experiment, the organic solvent was evaporated and the residue was dissolved in PPB buffer pH 7.0 solution. The solutions were filtered using membrane syringes (CA 0.45 μm) to remove large dust particles. The measurements were performed three times and averaged by the software. The morphology of the prepared surfaces was carried out by atomic force microscopy (Dimension Icon AFM system; Bruker, Billerica, MA, USA), and operated in tapping mode in the air. The pictures of AFM were visualized and analyzed using a Gwyddion 2.60 software.

### 2.3. Functionalization of GOx and Its Activity

This enzyme was modified with ferrocenecarboxaldehyde (FcCHO). The experiment was carried out in chloroform, in which the *K_M_* value of GOx was the closest to GOx in the PPB solution. FcCHO and GOx were taken in a molar ratio of 1000:1, i.e., 6.7 mg and 5.0 mg, respectively, and dissolved in 5 mL of chloroform. The mixtures were kept in an ultrasonic bath for several minutes and then stirred for 24 h at room temperature. After that chloroform was evaporated using a rotary evaporator at a bath temperature of 40 °C, the dry samples were dissolved in 5 mL of the PPB buffer. The solutions were filtered with membrane syringes (CA 0.45 μm). ROTI^®^Spin, MINI-30 centrifuge tubes with molecular weight cut-off of 30 kDa were used to collect the ferrocene-modified GOx fraction on the filter of the tube. Samples of 0.5 mL were added to the test tubes and centrifuged for 7 min at 8000 rpm. The process was repeated until the entire volume of the samples (5 mL) was filtered. The collected precipitates were additionally washed with 0.5 mL of PPB. The supernatant was discarded and the collected precipitates were dissolved in 5 mL of PPB and used in further experiments.

Activities of the unaffected and modified enzyme samples were measured via the aid of a coupled *o*-dianisidine-peroxidase reaction employing UV–Vis spectroscopy. The obtained initial velocity (*v*) vs. the glucose concentration curves were used to calculate the Michael constants (*K_M_*), and maximal rate (*v_max_*) values by applying the Michaelis–Menten Equation (1):(1)$$v=\frac{v_{max}\;C\left(GLU\right)}{K_{M}+C\left(GLU\right)}$$
where *C*(*GLU*) is a concentration of glucose in the sample. These kinetic parameters (*K_M_* and *v_max_*) for this reaction of the *o*-dianisidine oxidation were fitted by employing OriginPro 2015 software.

### 2.4. Immobilization of the Self-Assembly Monolayers of the Enzymes on Au

Gold electrodes were cleaned with Micropolish alumina 0.3 μm gel for 5 min, dipped in distilled water and stored in an ultrasonic bath for 5 min. Each electrode was cleaned by cyclovoltammetry (CV) in 50 mM KOH from 0 V to −2.6 V vs. Ag/AgCl using 40 scans at a scan rate of 300 mV s^−1^, and then in 0.5 M H_2_SO_4_ solutions from −0.2 V to 1.75 V vs. Ag/AgCl using 40 scans at a scan rate of 300 mV s^−1^. The Au electrodes were washed with distilled water and then immersed in Eppendorf tubes containing 5 mM 4-mercaptobenzoic acid (MBA) dissolved in methanol and stored for one hour in a refrigerator at +4 °C. Subsequently, these Au/MBA electrodes were transferred to a tube containing a mixture of EDC and NHS solutions with concentrations of 2 mM and 5 mM, respectively. The Au/MBA electrodes were kept in this mixture for 30 min at room temperature. After this step, the electrodes were immersed in the Fc-GOx solutions (1 mg mL^−1^) in a buffer pH 7.0 solution. The electrode in this solution was tightly wrapped with parafilm tape and stored for 12 h in a refrigerator at +4 °C. For comparison, two control Au/MBA and Au/MBA/GOx electrodes were prepared under the same conditions but using only MBA and the native GOx, respectively.

### 2.5. Electrochemical Investigation of the Electrodes

For these CV measurements, a three-electrode system consisting of gold coated with the modified Fc-GOx or GOx monolayer (Fc-GOx or GOx/MBA/Au) electrode of a diameter of 3.0 mm as a working electrode, Pt wire of diameter of 1 mm as the counter and Ag/AgCl in saturated KCl as a reference electrode were used. They were immersed in 25 mL of PPB buffer pH 7.0 containing 0.1 M KCl to increase the electrical conductivity of the solution. For signal detection, a potential range from 0 mV to 600 mV was chosen at the scan rate of 20 mV s^−^^1^. Two scans were made per measurement and a second scan was used as the results. The experiments were performed by increasing the concentration of glucose *C*(*GLU*) until the signal stabilized (up to 30 mM). Similar measurements were made in an oxygen-free environment, i.e., the solution was purged with Ar gas for 15 min and the CV experiments were carried out under an Ar atmosphere. The limit of detection (LOD) for the corresponding electrode was calculated using the Equation (2):(2)$$\mathrm{LOD}=\frac{3.0\sigma }{a}$$where *a* is the slope and *σ* is a standard deviation of the linear curve and a signal to noise (S/N) ratio of 3.0 was used. These parameters were calculated using a Data analysis tool from MS 2007 Excel.

## 3. Results and Discussion

### 3.1. Functionalization of Glucose Oxidase with Ferrocenyl Group

To functionalize glucose oxidase (GOx) with a ferrocenyl group, ferrocenyl aldehyde as a hole mediator was used. However, this reagent is not soluble in water (PPB of pH 7.0 in our case). Therefore, this condensation reaction between free side amino groups of the lysine or arginine residues on the GOx surface and this aldehyde can only occur in organic solvents. Before performing the functionalization of the GOx surface, the enzyme stability after storage in chloroform as the organic solvent was tested using a coupled enzyme assay, in which GOx oxidizes glucose resulting in the production of H_2_O_2_ that reacts by catalyzing horseradish peroxidase (HRP) with *o*-dianisidine as the colorimetric probe at 460 nm. To estimate and compare Michaelis constant (*K_M_*) and maximal rate (*v_max_*) as the catalytic parameters of the affected and unaffected GOx, a Michaelis–Menten model was used. The values of *K_M_* and *v_max_* of these samples were measured to be 11.2 ± 0.8 mM^−1^, 6.0 ± 0.1 mM s^−1^, and 13.7 ± 1.6 mM^−1^, 5.2 ± 0.2 mM s^−1^ for unaffected GOx and after its exposure in CHCl_3_, respectively. These results show that the activity after GOx exposure in CHCl_3_ is lower but retains its catalytic properties. Therefore, this organic solvent can be used to functionalize this enzyme.

Compounds with the active aldehyde function groups have been used to functionalize the N-terminus residues of amino acids and for immobilization of enzymes onto electrode surfaces while maintaining the activity of these enzymes [11]. Therefore, ferrocenecarboxaldehyde (FcCHO) as a redox-responsive reagent was used to functionalize the surface of GOx. The mixture of FcCHO and GOx in a molar ratio of 1000:1 in chloroform was stirred at room temperature for 24 h. After this reaction, the organic solvent was evaporated and the residue was dissolved in PPB of pH 7.0. For purification of the functionalized Fc-GOx, a centrifuge tube with a molecular weight cut-off of 30 kDa was used to collect the Fc-GOx fraction on its filter. The Fc-GOx fraction was dissolved again in the buffer solution and its activity was tested using the coupled enzyme assay. The kinetic parameters of *K_M_* and *v_max_* were measured to be 17.0 ± 1.1 mM and 4.5 ± 0.2 mM s^−1^ for Fc-GOx, respectively. The activity of Fc-GOx is lower than the unaffected GOx but Fc-GOx retains the catalytic properties.

### 3.2. Characterization of the Functionalized Enzyme

The optical UV–Vis and CD spectroscopy methods were carried out to characterize the obtained Fc-GOx enzyme and these results were compared with initial and GOx after its exposure to CHCl_3_. The UV–Vis bands at around 330 nm and 450 nm should be characteristic of ferrocenyl aldehyde and its derivatives [18]. However, we observed only the brood band at around 320 nm in the UV–Vis spectrum in a logarithmic scale of Fc-GOx, which can be attributed to the ferrocenyl group (Figure 1b). From the UV–Vis spectra (Figure 1b), the bands at 280 nm and 207 nm, can be attributed to an *n* → π* transition in the aromatic amino acid moieties, with a charge transition from the flavin adenine dinucleotide (FAD) as the cofactor to indolyl moieties of tryptophane in the active center of GOx, and the full π → π* transition between the chain of the aromatic residues of the enzyme, respectively [19]. These UV–Vis bands of Fc-GOx are widespread and this phenomenon is characteristic of aggregated peptides [20]. A similar widespread spectrum after GOx exposure in CHCl_3_ was observed (Figure 1b). To investigate the structural changes after the functionalization of this enzyme, circular dichroism (CD) spectra of these samples were measured (Figure 1a). We indicated that the intensity of the corresponding CD bands of Fc-GOx at 210 nm and 280 nm are decreased. This indicates that the positions of FAD and tryptophan residues in the active center of Fc-GOx are changed during chemical functionalization or after GOx exposure to CHCl_3_ of this enzyme. However, the enzyme retains its tertiary structure.

Based on the observation from the UV–Vis spectrum that the enzyme can aggregate to nanoparticles, hydrodynamic diameters of the Fc-GOx, unaffected GOx, GOx after CHCl_3_ samples in the buffer solution were measured using a dynamic light scattering (DLS) method. Figure 1c shows that particles with diameters of 10 nm and a small fraction with a diameter of ~200 nm were recorded for the unaffected GOx in the buffer solution. This particle size of 10 nm well corresponds to the diameter of the starting enzyme. However, Figure 1c shows that nanoparticles of a diameter of 280 nm are formed after GOx exposure to CHCl_3_. Moreover, the two fractions of nanoparticles with diameters of around 10 nm and 160 nm are characteristic of the Fc-GOx sample (Figure 1c). These results clearly confirm that the enzyme aggregated to nanoparticles under the action of this organic solvent. We plan to investigate this interesting effect in our next work. It is worth mentioning that the hydrodynamic diameter of the functionalized enzyme did not change after this chemical manipulation.

### 3.3. Self-Assembly Monolayers of GOx and Fc-GOx on Au

Self-assembled molecular monolayers (SAMs) of DET-able enzymes have been applied as the best method for immobilization of these enzymes on the electrode surface because the tunneling distance is a key factor in DET-type bioelectrocatalysis [21]. Therefore, the GOx and Fc-GOx enzymes were immobilized by a step-by-step method on Au electrodes as the SAMs. First of all, 4-mercaptobenzoic acid (MBA) was used to deposit SAM on the surface of the Au electrodes (Au/MBA). Secondly, carboxylic acid groups of this acid on the Au/MBA surface were activated using 1-ethyl-3-(3-dimethylaminopropyl)carbodiimide (EDC) and *N*-hydroxysuccinimide (NHS) as the classical amide bond forming and the immobilization reagents. These Au/MBA electrodes activated with the NHS groups were immersed into corresponding GOx or Fc-GOx solutions to obtain the SAMs of the enzymes (GOx/MBA/Au and Fc-GOx/MBA/Au) on the Au surface (Figure 2a).

To characterize the GOx/MBA/Au and Fc-GOx/MBA/Au surfaces, atomic force microscopy (AFM) was used. Figure 2b,c shows the GOx/MBA/Au and Fc-GOx/MBA/Au surfaces at the dimensions of 10 × 10 μm. The roughness parameters, such as average roughness (R_a_), root mean square roughness (R_q_) and maximum elevation were estimated by analyzing topography scans of these surfaces. The parameters of R_a_, R_q_, and maximum elevation were calculated to be 0.83 nm, 1.0 nm, and 7.4 nm for GOx/MBA/Au, 3.4 nm, 5.1 nm, and 43 nm for Fc-GOx/MBA/Au, and for comparison 0.66 nm, 0.93 nm, and 2.5 nm for bare Au, respectively. These AFM results demonstrate that the surface of GOx/MBA/Au is much less rough than Fc-GOx/MBA/Au because of structures, such as the aggregated enzymes with diameters of ~200 nm and highs of ~40 nm were detected on the Fc-GOx/MBA/Au surface. However, these structures on the surface are rare due to the relatively low concentration of nanoparticles in the solution used for immobilization. After AFM at the higher resolution of 200 × 200 nm, grains with diameters of ~10 nm and highs of ~2 nm were observed (Figure 2d). These grains were not indicated on the bare Au surface at the same resolution (Figure 2e). The distribution of these grain diameters shows that the most common diameters are in an interval of 10–12 nm (Figure 2f). These results fit perfectly with our data from the dynamic light scattering experiments (Figure 1c) and from the dimensions of the crystal structure of this enzyme [22]. These results are good proof that the monolayer of GOx is deposited on the surface. Moreover, 140 grains of the enzymes were counted on the Fc-GOx/MBA/Au sample with an area of 4 × 10^−10^ cm^2^ (200 × 200 nm). The concentration of the GOx molecules on the surface was calculated to be 3.5 × 10^11^ molecules cm^−2^ or 5.8 × 10^−13^ mol cm^−2^.

### 3.4. Bioelectrocatalysis of GOx/MBA/Au and Fc-GOx/MBA/Au with and without Oxygen

To assess the suitability of the modified enzyme for use in biosensors, the electrode with the deposited Fc-GOx/MBA/Au monolayer was investigated and compared with the GOx/MBA/Au monolayer-based electrode. Figure 3a shows that a broad oxidation signal at around 0.4 V vs. Ag/AgCl, which also is characteristic for starting FcCHO in the inset of Figure 3a, was observed. However, this electrochemical signal was not observed from the CV of the GOx/MBA/Au and MBA/Au monolayers. Therefore, this signal attributes to oxidation of the Fc group on the GOx surface. To show the effect of oxygen on the performance of biosensors, these Fc-GOx/MBA/Au and GOx/MBA/Au electrodes were electrochemically studied with and without oxygen under varying glucose concentrations in the electrolyte (Figure 3b–e), respectively.

To show the bioelectrocatalytic properties of the electrodes, responses of the current densities to the concentrations of glucose at 300 mV vs. Ag/AgCl were depicted in Figure 3d,h. On the one hand, Figure 3d,h shows that the best bioelectrocatalytic properties are obtained using the Fc-GOx/MBA/Au electrode without oxygen. The Fc-GOx/MBA/Au electrode exhibits a greater response in the argon-saturated than in an aerated buffer. However, the corresponding electrode with native GOx (GOx/MBA/Au) without oxygen demonstrated a poor response to these properties. On the other hand, the Fc-GOx/MBA/Au electrode at low glucose concentrations with oxygen demonstrated these properties well. Moreover, based on linearity at 300 mV vs. Ag/AgCl of the Lineweaver–Burk’s test (Figure 3i), the curves at the low glucose concentrations (i.e., at higher values of *C*(*GLU*)^−1^) for Fc-GOx/MBA/Au with and without oxygen are almost parallel. Therefore, the bioelectrocatalytic characteristics are the same with each other for Fc-GOx/MBA/Au with and without oxygen. Therefore, oxygen is not involved in this catalytic reaction when we used the functionalized GOx. However, a lower glucose concentration than around 0.1–0.2 mM cannot be detected by using the GOx/MBA/Au electrode. These values are equal to the concentration of oxygen in an air-saturated aqueous solution of ~0.2 mM. Similar results for carbazole derivatives as the hole transporting semiconductors coated electrodes were observed in our previous works [8,9]. Therefore, the bioelectrocatalytic properties of this electrode depend on the oxygen concentration. Moreover, the starting MBA/Au electrode did not demonstrate these responses (Figure 3g,h).

To obtain the performance characteristics of the biosensors based on the Fc-GOx/MBA/Au electrode without and with oxygen, linear ranges, average sensitivities, and limit of detection (LOD) at the applied potential of 300 mV vs. Ag/AgCl were calculated and summarized in Table 1. The linear ranges from 20 μM to 80 μM for Fc-GOx/MBA/Au at both conditions were measured (Table 1, entries 1 and 2). However, these linear ranges for GOx/MBA/Au are at the higher concentrations, i.e., from 1.0 mM to 5.0 mM (Table 1, entries 3 and 4). The good average sensitivities for Fc-GOx/MBA/Au were calculated based on the corresponding linear ranges to be 2.33 μA cm^−2^ mM^−1^ and 1.40 μA cm^−2^ mM^−1^ in the argon-saturated and aerated buffer, respectively. However, the lower values of the average sensitivities were estimated for GOx/MBA/Au under these conditions (Table 1, entries 3 and 4). Thus, the bioelectrocatalytic reaction of glucose oxidation occurred approximately 170 times faster using the Fc-GOx/MBA/Au than GOx/MBA/Au electrode without oxygen. Moreover, this kind of biosensor using Fc-GOx had the low LODs of 5.2 μM and 8.3 μM without and with oxygen, respectively. However, much higher LODs using GOx/MBA/Au with and without oxygen were estimated to be 210 μM and 38 μM, respectively. These values of LODs show that lower concentrations than 0.2 mM cannot be determined using GOx/MBA/Au due to oxygen competition with the bioelectrocatalytic reaction.

These obtained characteristics of glucose sensing of the electrodes (Table 1, entries 1–4) were compared with other electrodes based on Fc and GOx (Table 1, entries 5–8) and hole transporting organic semiconductors (Table 1, entries 9 and 10) in the literature. Moreover, direct electrical communication between chemically modified enzymes with ferrocene derivatives and metal electrodes has been reported by Degani and Heller [17], and Bartlett et al. [15]. However, the authors did not provide the characteristics of these biosensors. On the other hand, ferrocene-based conducting polymers deposited GOx onto the electrode in order to obtain surfaces with the mediating properties. Many authors have chosen this strategy for the preparation of biosensors (Table 1, entries 5–8). These characteristics are compared with our results in Table 1. Yuan et al. have demonstrated that gold nanoparticles decorated with GOx and 6-(ferrocenyl)hexanethiol on a single layer graphene electrode can be applied for electrochemical ultrasensitive glucose biosensor (Table 1, entry 8) [26]. Therefore, surface nanostructurization can be a great way to reduce the level of detection. In addition, more data can be found in the review [10]. Our results and the literature data have been compared with electrodes based on polycarbazoles as the hole transporting organic semiconductors (Table 1, entries 9 and 10). However, these electrodes do not show values of LOD less than 0.1–0.2 mM. Synthesis of corresponding monomers and several laborious steps to introduce the Fc groups onto the electrode surface have been required for these electrode modification procedures. In this work, we have proposed a simpler method without genetic engineering methods for modifying enzymes in an organic solvent.

### 3.5. Possible Mechanism

In this work, we show that electrical communication between a cofactor as the redox center of immobilized GOx and a gold electrode can be established by the chemical modification of the enzyme. This process is predominant in the absence of oxygen in the solution. In addition, we have previously predicted that hole hopping between the reduced cofactor and electrode surface through aromatic redoxable amino acid residues is possible [5]. The possible mechanism is shown in Figure 1. First of all, glucose reduces FAD to FADH_2_. We have previously shown that direct hole and electron tunneling from the cofactor to the electrode (or vice versa) is not possible due to the long distance between the cofactor and the electrode surface. Secondly, the reduced cofactor regenerates by holes (H^+^) as the positive charge carriers from the side aromatic redoxable amino acid residues. These charge carriers come through these residues via the hole hopping mechanism from the mediator, which is nearest to the surface of the electrode. Finally, electron relays from the oxidized mediator (i.e., radical cation of Fc) to the Au surface via the electron tunneling mechanism. Thus, an increase in the biocatalytic current is observed. Although without oxygen, this mechanism dominates, in absence of oxygen it competes with the classic mechanism, in which FADH_2_ is oxidized by oxygen to release H_2_O_2_. This reduced form of the cofactor can be oxidized by oxygen, so most biosensors of this type cannot detect concentrations below ~0.2 mM, because it is the concentration of dissolved oxygen in the aqueous solution. We have achieved that cofactor electrooxidation occurred without oxygen competition. Therefore, the search for new materials to help transfer holes from the enzyme surface to the cofactor is an important challenge.

## 4. Conclusions

In summary, chloroform is an excellent organic solvent to functionalize glucose oxidase with ferrocenyl as the redoxable groups. Using a coupled enzyme assay, we found that the activity of the functionalized enzyme (Fc-GOx) was lower than the unaffected GOx but Fc-GOx retained its catalytic properties. After our chemical manipulation in this organic solvent, diameters of this functionalized enzyme of ~10 nm and ~160 nm were observed from the DLS experiments, which correspond to individual enzymes and their aggregates caused by this organic solvent, respectively. The self-assembly monolayer of Fc-GOx was formed and characterized by AFM. These results have demonstrated the formation of a monolayer from the single-molecule enzyme layers on the gold electrode. Moreover, the concentration of the GOx molecules on this surface was calculated to be 3.5 × 10^11^ molecules cm^−2^. The bioelectrocalatytic properties of the GOx and Fc-GOx-based monolayer-based electrodes were electrochemically investigated and utilized for the detection of glucose. The linear ranges from 20 μM to 80 μM with LODs of 5.2 μM and 8.3 μM (S/N = 3) for the Fc-GOx-based electrode were found in the argon-saturated and aerated buffer, respectively. However, glucose concentrations lower than ~0.2 mM cannot be detected by using the unaffected GOx-based electrode due to oxygen competition with the bioelectrocatalytic reaction. We have successfully demonstrated that the enzyme can be functionalized with the ferrocenyl groups in chloroform and applied for the detection of glucose via the quasi-DET mechanism. This quasi-DET ability of the functionalized GOx provides an opportunity to construct a mediator-free type glucose biosensor.

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
