# Peer review of "Functionalization of Glucose Oxidase in Organic Solvent: Towards Direct Electrical Communication across Enzyme-Electrode Interface"

_biosensors, 2022, doi:10.3390/bios12050335_

Round 1

Reviewer 1 Report

The authors reported a simple method to functionalize glucose oxidase with redox-able ferrocene groups in chloroform, which then was employed to develop the biosensing monolayer-based platforms for the detection of glucose utilizing the quasi-DET mechanism. The results revealed that the formation of harmful H2O2 is minimized during enzymatic electrocatalysis due to a electrochemical regeneration of the catalytic center. The characterization and testing are relatively comprehensive. Thus, I recommend this manuscript can be accepted for publication in this journal after addressing the following minor issues.

  1. The resolution of Figure 2f is too low so that corresponding information cannot be clearly identified. Please provide a high-resolution Figure 2f.
  2. An appropriate distance should be kept between Figure 3e and 3f.
  3. Can the authors provide more literatures for comparison in Table 1 s to better demonstrate the superiority of fabricated sensors in this manuscript.
  4. Except the harmful aspect from hydrogen peroxide in some specific experiments, as an important chemical product, some useful or important aspects regarding with hydrogen peroxide are suggested to provide in the manuscript, and references such as https://doi.org/10.1002/anie.202200086 and  https://doi.org/10.1002/ange.202006747 can be checked by authors.

Author Response

Dear Editor and Referees,

We are pleased to submit the revised research article entitled “Functionalization of Glucose Oxidase in Organic Solvent: Towards Direct Electrical Communication across Enzyme-Electrode Interface” (Manuscript ID: biosensors-1696347) and answers to the Referees’ comments and questions.

We are truly grateful to critical comments and thoughtful suggestions from the Referees. Based on these comments and suggestions, we have made careful changes highlighted in the revised manuscript. All the changes made to the text are marked in yellow colour.

Thank you for your consideration of our manuscript. We look forward to hearing from you.

Sincerely,

Dr Gintautas Bagdžiūnas

Reviewer 2 Report

The manuscript entitled “Functionalization of Glucose Oxidase in Organic Solvent: Towards Direct Electrical Communication across Enzyme-Electrode Interface” by VygailÄ— et al. reports the enzyme functionalization in organic solvent towards glucose sensor. However, as research work, it still suffers some deficiencies. The manuscript at its current form is not suitable for acceptance for Biosensors Journal. The following points should be carefully addressed before resubmission;

  1. Please check the format of references.
  2. The authors mentioned that “Therefore, this condensation reaction between free side amino groups of the lysine or arginine residues on the GOx surface and this aldehyde can only occur in organic solvents.”. Except for UV-Vis and DLS methods that were reported, please provide further results to confirm the reaction occurs between functional groups (e.g. FT-IR, XPS, etc.)
  3. Through AFM high resolution image of 200×200 nm (Fig. 2f), the most common diameters of Fc-GOx/MBA/Au are an interval 10 – 12 nm. But, as shown in Fig. 1c, the nanoparticles with diameters are around 10 nm and 160 nm. Why are the Fc-GOx compounds with a large diameter (160 nm) not observed by AFM? Please further explain in detail.
  4. Why is Fc-GOx/MBA/Au electrode not depend on the oxygen concentration? Please explain more details.
  5. Please provide the results in real samples.
  6. Please show the stability graph of Fc-Gox/MBA/Au electrode.

Author Response

(The authors gave the same response as above.)

Round 2

Reviewer 2 Report

The manuscript entitled “Functionalization of Glucose Oxidase in Organic Solvent: Towards Direct Electrical Communication across Enzyme-Electrode Interface” by VygailÄ— et al. reports the enzyme functionalization in organic solvent towards glucose sensor. After major revision, the manuscript at its current form is suitable for acceptance for Biosensors Journal.